# METRIC LEARNING ON TEMPORAL GRAPHS VIA FEW-SHOT EXAMPLES

## ABSTRACT

Graph metric learning methods aim to learn the distance metric over graphs such that similar graphs are closer and dissimilar graphs are farther apart. This is of critical importance in many graph classification applications such as drug discovery and epidemics categorization. In many real-world applications, the graphs are typically evolving over time; labeling graph data is usually expensive and also requires background knowledge. However, state-of-the-art graph metric learning techniques consider the input graph as static, and largely ignore the intrinsic dynamics of temporal graphs; Furthermore, most of these techniques require abundant labeled examples for training in the representation learning process. To address the two aforementioned problems, we wish to learn a distance metric only over fewer temporal graphs, which metric could not only help accurately categorize seen temporal graphs but also be adapted smoothly to unseen temporal graphs. In this paper, we first propose the streaming-snapshot model to describe temporal graphs on different time scales. Then we propose the METATAG framework: 1) to learn the metric over a limited number of streaming-snapshot modeled temporal graphs, 2) and adapt the learned metric to unseen temporal graphs via a few examples. Finally, we demonstrate the performance of METATAG in comparison with state-of-the-art algorithms for temporal graph classification problems.

## 1 INTRODUCTION

Metric learning aims to learn a proper distance metric among data items in the input space, which reflects their underlying relationship. With the prevalence of graph data in many real-world applications, it is of key importance to design a good distance metric function for graph data, such that the output value of the function is small for similar graphs and large for dissimilar ones. Many downstream tasks on the graph data can benefit from such a distance metric. For example, it could lead to significantly improved classification accuracy for graph classification in many domains such as protein and drug discovery (Schölkopf et al., 2004; Dai et al., 2016), molecular property prediction (Duvenaud et al., 2015; Gilmer et al., 2017), and epidemic infectious pattern analysis (Derr et al., 2020; Oettershagen et al., 2020); it could also speed up the labeling of graph data in an active learning framework (Macskassy, 2009).

However, current graph metric learning methods (Shaw et al., 2011; Tsitsulin et al., 2018; Bai et al., 2019; Li et al., 2019; Yoshida et al., 2019) assume the input graph data as static and ignore evolution patterns of temporal graphs, which may also provide insights for identifying the graph property (Isella et al., 2011). To best of our knowledge, there is currently no algorithm designed for learning metrics over temporal graphs to further involve evolution pattern consideration into the learned metric space. On the other hand, facing limited i.i.d. data, traditional metric learning methods (Goldberger et al., 2004; Salakhutdinov & Hinton, 2007) have been extended to the few-shot learning by transferring the learned metric across different tasks (Vinyals et al., 2016; Snell et al., 2017; Oreshkin et al., 2018; Allen et al., 2019). Label scarcity problem also occurs in the graph research community, because labeling graph data is typically expensive and requires background knowledge (Hu et al., 2020a;b; Qiu et al., 2020), especially for domain-specific applications such as biological graph data labeling (Zitnik et al., 2018). Inspired by that, graph metric learning via few-shot examples has recently attracted many nascent researchers' attention. But, the majority has been devoted to the node-level metric learning (Yao et al., 2020; Suo et al., 2020; Huang & Zitnik, 2020; Lan et al., 2020; Wang et al.,

2020; Ding et al., 2020), only a few nascent efforts focus on the graph-level metrics (Ma et al., 2020; Chauhan et al., 2020), and all of them ignore the graph dynamics but take static graphs as input.

To wrap up, these discussed-above observations bring three bottlenecks to present temporal graph metric learning algorithms: 1) How to learn a good metric over temporal graphs, especially on the entire graph level (i.e., accuracy of metrics); 2) How to ensure the learning process only consumes less labelled temporal graph data; and 3) How to smoothly apply that learned metric to identify unseen graphs (i.e., flexibility of metrics). In this paper, we wish to learn a distance metric only over fewer temporal graphs, which metric (as shown in Figure 1) could not only help accurately classify seen temporal graphs during each metric learning task, but also be adapted smoothly to new metric learning tasks and converge fast (i.e., several training iterations) to classify unseen temporal graphs by consuming a few labeled examples.

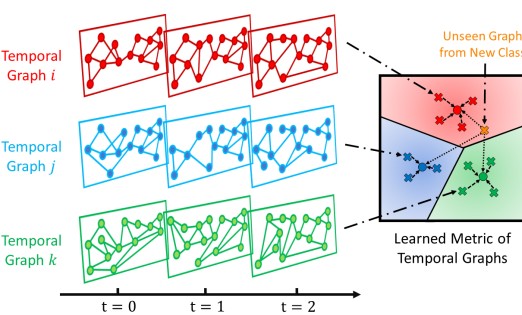

Figure 1: An example of metric learning on several temporal graphs. In the right box, each 'x' denotes a temporal graph representation, and each circle denotes a class representation.

Our main contributions can be summarized as:

- To describe the evolving graph in a fine-grained manner, we propose the *streaming-snapshot model* that contains multiple time scales suitable for complex real-world scenarios and other merits are discussed in Section 3.
- To learn the metric over a bunch of streaming-snapshot modelled temporal graphs, we propose the *prototypical temporal graph encoder* to extract the lifelong evolution representation of a temporal graph with the proposed *multi-scale time attention mechanism*, such that temporal graphs from the same class share the similar encoded patterns; To make the extracted metric rapidly adapt to unseen temporal graphs with only a few examples, we introduce a *meta-learner* to transfer and tailor knowledge and encapsulate it with the *prototypical temporal graph encoder* into an end-to-end model, called METATAG.
- We conduct the temporal graph classification experiments on biological network domain and social network domain, which show the effectiveness of METATAG compared with state-of-the-art algorithms. Also, we analyze the convergence speed of METATAG during the meta-testing, the parameter sensitivity, and the ablation study of each part of METATAG.

## 2 PRELIMINARIES

**Graph Metric Learning**. Learning a distance metric is closely related to the feature extraction problem (Globerson & Roweis, 2005; Salakhutdinov & Hinton, 2007). To be specific, given any distance metric $\mathbb{D}$, we can measure distance $\mathbb{D}(\mathbf{x}_i, \mathbf{x}_j)$ between two input feature vectors $\mathbf{x}_i \in \mathbb{R}^m$ and $\mathbf{x}_j \in \mathbb{R}^m$ by computing $\mathbb{D}'(f_\theta(\mathbf{x}_i), f_\theta(\mathbf{x}_i))$, where $f_\theta$ is a learnable function mapping the input feature $\mathbf{x}_i \in \mathbb{R}^m$ into the latent feature $\mathbf{h}_i = f_\theta(\mathbf{x}_i) \in \mathbb{R}^f$ (Salakhutdinov & Hinton, 2007). The transformation function $f_\theta$ could be linear or non-linear (Wang & Sun, 2015). When $f_\theta$ is a linear function $f_\theta(\mathbf{x}_i) = \mathbf{W}\mathbf{x}_i$, learning a generalized Mahalanobis metric $\mathbb{D}$ can be expressed as follows.

$$
\begin{aligned}
\mathbb{D}(\mathbf{x}_i, \mathbf{y}_j) &= \sqrt{(\mathbf{x}_i - \mathbf{x}_j)^\top \mathbf{M}(\mathbf{x}_i - \mathbf{x}_j)} \\
&= \sqrt{(\mathbf{x}_i - \mathbf{x}_j)^\top \mathbf{W}^\top \mathbf{W}(\mathbf{x}_i - \mathbf{x}_j)} \\
&= \sqrt{(\mathbf{W}\mathbf{x}_i - \mathbf{W}\mathbf{x}_j)^\top (\mathbf{W}\mathbf{x}_i - \mathbf{W}\mathbf{x}_j)} \\
&= \mathbb{D}'(f_\theta(\mathbf{x}_i), f_\theta(\mathbf{x}_j))
\end{aligned}
\tag{1}
$$

where $\mathbf{M}$ is some arbitrary positive semi-definite matrix to be determined for the Mahalanobis metric $\mathbb{D}$, and $\mathbf{M}$ can be decomposed as $\mathbf{M} = \mathbf{W}^\top \mathbf{W}$. Then the Mahalanobis metric $\mathbb{D}$ on the input feature space is equivalent to the Euclidean metric $\mathbb{D}'$ on the hidden feature space, such that

learning a undetermined metric $\mathbb{D}$ (e.g., Mahalanobis) on input feature is equivalent to learning hidden features on a fixed metric $\mathbb{D}'$ (e.g., Euclidean) (Globerson & Roweis, 2005; Salakhutdinov & Hinton, 2007; Wang & Sun, 2015; Snell et al., 2017). Also, $f_\theta$ can be a non-linear transformation for involving more parameters to model higher-order correlations between input data dimensions than linear transformations (Salakhutdinov & Hinton, 2007; Wang & Sun, 2015; Snell et al., 2017). Based on the above analysis, we are ready to model our graph metric learning problem: learning a "good" distance metric over pairs of graphs is to learn a "good" mapping function $f_\theta$ of graphs in Euclidean space. The "goodness" is controlled by $\theta$ and we discuss how we define it in Section 3.

## 3 STREAMING-SNAPSHOT MODEL AND PROBLEM SETUP

The table of symbols is summarized in Appendix. To specify, we use bold lowercase letters to denote column vectors (e.g. $\mathbf{a}$), bold capital letters to denote matrices (e.g., $\mathbf{A}$), and $\mathbf{A}(i, :)$ to denote the $i$-th row of matrix $\mathbf{A}$. Also, we let the parenthesized superscript denote the timestamp like $\mathbf{A}^{(t)}$. We use graph and network interchangeably in this paper.

**Streaming-Snapshot Model**. In the streaming-snapshot model, there exists two kinds of timestamps, $t_e \in \{0, 1, \ldots, T_e\}$ denotes the edge timestamp and $t_s \in \{0, 1, \ldots, T_s\}$ denotes the snapshot timestamp. To be specific, we describe a temporal graph $\mathcal{G}$ as a sequence of timestamped snapshots $\{\mathcal{S}^{(t_s)}\}_{t_s=0}^{T_s}$, and each timestamped snapshot has a set of timestamped edges labeled as $(v_i, v_j, t_e, t_s)$. Note that, these two timestamps are different measures, they do not need to have the comparison relationship. In Figure 2, we provide a temporal graph example whose $T_e = 4$ and $T_s = 2$.

The merits of describing the temporal graph within the streaming-snapshot model include: 1) Carrying multi-scale complex temporal information. Some social networks change rapidly in the microscopic view (Leskovec et al., 2008), while some graphs like yeast metabolic graph (Tu et al., 2005) and repeating frames in video analysis (Li et al., 2020) change slowly in the macroscopic view (Leskovec et al., 2005). If the input temporal graph has these two evolution patterns (i.e., edge timestamps and snapshot timestamps), our streaming-snapshot model could handle both of them simultaneously because streaming model could describe the interaction graph in a rapid and continuous manner and snapshots could compensate for the complement by modeling episodic, slowly-changing, and periodical patterns (Aggarwal & Subbian, 2014). If not, our streaming-snapshot is also viable by downgrading into a single streaming or a single snapshot model. 2) Saving computation memory. When we need to generate the graph-level embedding for a long lifetime temporal graph, we only need to load each snapshot embedding vector instead of loading every node embedding that appears in the whole temporal graph. (The detail of how to generate a snapshot embedding through its relevant node embeddings is discussed in Section 4.1.1, i.e., Multi-Scale Time Attention Mechanism.) Beyond recent temporal graph representation learning methods (Pareja et al., 2020; Xu et al., 2020; Beladev et al., 2020) that only focus on one time scale and ignore the whole lifetime evolution representation, our method can learn the lifelong evolution pattern of a temporal graph on different time scales.

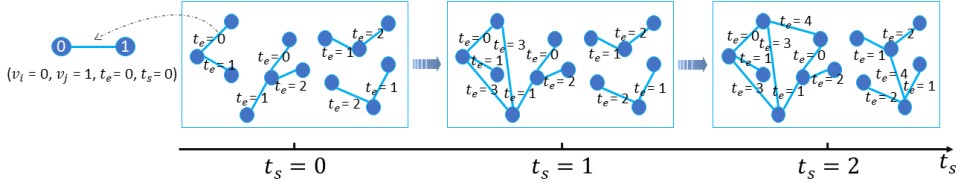

Figure 2: An example of a temporal graph described by the proposed streaming-snapshot model. Each edge is labeled by two timestamps, i.e., $(v_i, v_j, t_e, t_s)$, $t_e \in \{0, 1, 2, 3, 4\}$, and $t_s \in \{0, 1, 2\}$

As for the data structure, we store each edge as $(v_i, v_j, t_e)$ and each snapshot adjacency matrix as $\mathbf{A}^{(t_s)} \in \mathbb{R}^{|V^{(t_s)}| \times |V^{(t_s)}|}$, i.e., $V^{(t_s)} \subseteq V$ and $|V^{(t_s)}| \neq |V^{(t_s+1)}|$ is allowable. Although our method is readily designed for evolving input features according to different timestamps, for the notation clarity, we denote the node feature matrix $\mathbf{X} \in \mathbb{R}^{n \times m}$, such that the input node feature of temporal graph $\mathcal{G}$ is already time-aware, and $n = |V|$ and $m$ denotes the dimension of features.

**Problem Setup**. With the streaming-snapshot modelled temporal graphs, our goal is to learn a parameterized metric that could accurately classify seen temporal graphs and also be smoothly

adapted to unseen temporal graphs. Based on above analysis, this problem can be solved by learning a "good" graph representation learning function $f_\theta$ in Euclidean metric. To further achieve this "goodness" only with less labelled data, we formalize $f_\theta$ into a bi-level meta-learning paradigm (Finn et al., 2017). Given the streaming-snapshot modelled temporal graphs and corresponding labels $\widetilde{\mathcal{G}} = \{(\mathcal{G}_0, y_0), (\mathcal{G}_1, y_1), \ldots, (\mathcal{G}_n, y_n)\}$, we split $\widetilde{\mathcal{G}}$ into $\widetilde{\mathcal{G}}^{train}$ for meta-training and $\widetilde{\mathcal{G}}^{test}$ for meta-testing, where the testing set only has unseen graph labels from the training set. We shuffle the training set $\widetilde{\mathcal{G}}^{train}$ to sample graph metric learning tasks following a distribution $\mathcal{T}_i \sim P(\mathcal{T})$, where each graph metric learning task $\mathcal{T}_i$ is realized by a $K$-way $N$-shot temporal graph classification task based on the graph representation $f_{\theta_i}(\mathcal{G}_n)$. During each task $\mathcal{T}_i$, we sample a support set $\widetilde{\mathcal{G}}^{train}_{support}$ and a query set $\widetilde{\mathcal{G}}^{train}_{query}$, such that the support set is used to train the graph representation function $f_{\theta_i}$ to accurately predict the graph labels of the query set. At the meta-testing stage, we transfer the learned knowledge from each task (i.e., $\theta_i$) to the meta-learner (i.e., $\Theta$), then we update $\Theta$ a few times by classifying unseen temporal graphs on support set $\widetilde{\mathcal{G}}^{test}_{support}$, finally we report the classification accuracy of fine-tuned $\Theta$ on query set $\widetilde{\mathcal{G}}^{test}_{query}$. The concrete objective and loss function of each graph metric learning task $\mathcal{T}_i$, i.e., the "goodness", is mathematically expressed in Section 4.

## 4 METATAG FRAMEWORK

We illustrate METATAG with the proposed *prototypical temporal graph encoder* and *meta-learner*. First, prototypical temporal graph encoder captures temporal graph lifelong evolution representations through the *multi-scale time attention mechanism*, which serves for learning the parameterized metric (i.e., $\theta_i$) in each graph metric learning task (i.e., $\mathcal{T}_i$). Second, meta-learner $\Theta$ transfers the knowledge $\theta_i$ learned from each task $\mathcal{T}_i$ for the fast adaption on unseen temporal graphs classifications.

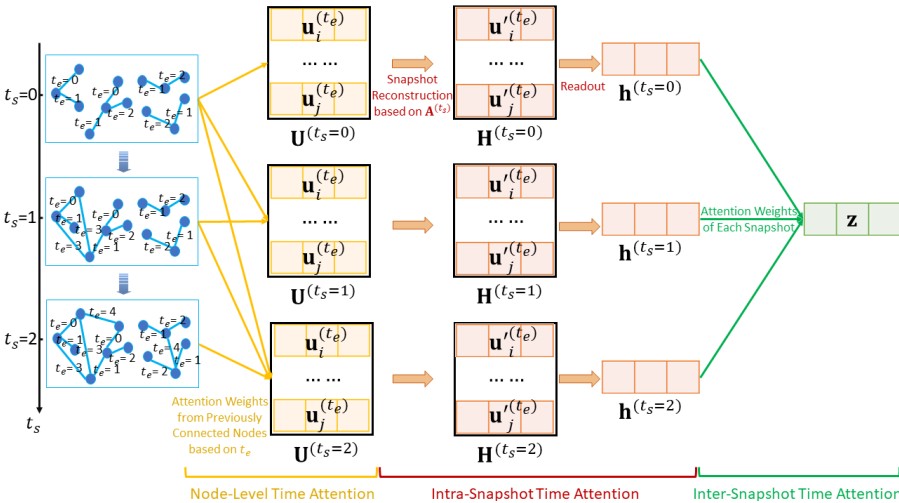

Figure 3: The multi-scale time attention mechanism of the prototypical temporal graph encoder for encoding temporal graph representation vector $\mathbf{z}$ of input temporal graph $\mathcal{G}$.

## 4.1 PROTOTYPICAL TEMPORAL GRAPH ENCODER

As stated in Eq. 1, the metric $\mathbb{D}$ is pair-wise. To save storage and computation complexity, learning $\mathbb{D}$ with labels involves the class representation concept (e.g., chunklet in (Bar-Hillel et al., 2005) and prototype in (Snell et al., 2017)), such that a sample should be close to its class representation and far from other class representations in metric $\mathbb{D}$. In this paper, we follow (Snell et al., 2017) to name class representation as prototype. In each graph metric learning task $\mathcal{T}_i$, prototypical temporal graph encoder encodes class-distinctive lifelong evolution patterns within two steps. First, multi-scale time attention mechanism is responsible for learning a single temporal graph representation from different time domains, i.e., edge timestamps and snapshot timestamps. Second, prototype generator generates the prototype for the same class temporal graph representations, to ensure same class graphs share the same prototypical pattern.

---

**Algorithm 1** Sample Time-Aware Adjacent Node Sequence $N_v^{(t_e)}$ for Node $v$ at Edge Timestamp $t_e$

---

**Input:** node $v$ at $t_e$, temporal graph $\mathcal{G}$
1: **for** edge timestamp $t < t_e$ **do**
2:    **if** edge $(v', v, t)$ exists **then**                                    ▷ connected edges before time $t_e$
3:       $N_v^{(t_e)}$ appends $\mathbf{X}(v', :)\|\mathcal{K}(t, t_e)$    ▷ concatenation of $\mathbf{X}(v', :) \in \mathbb{R}^m$ and $\mathcal{K}(t, t_e) \in \mathbb{R}^d$
4:    **end if**
5: **end for**

---

#### 4.1.1 MULTI-SCALE TIME ATTENTION MECHANISM

As shown in Figure 3, multi-scale time attention mechanism encodes the streaming-snapshot modeled temporal graph $\mathcal{G}$ into the representation vector $\mathbf{z}$ through three components, i.e., node-level time attention, intra-snapshot time attention, and inter-snapshot time attention.

**Node-Level Time Attention**. Intuitively, this mechanism first uses edge timestamp $t_e$ to learn streaming pattern of the input graph. We first need to sample a time-aware adjacent node sequence $N_v$ for each node $v$ in the temporal graph $\mathcal{G}$ (as shown in Alg. 1), then we apply the node-level time attention mechanism to learn the time-aware embedding $\mathbf{u}_v$ of node $v$ from its previous connected nodes. However, self-attention mechanism has become the key component for representing sequential data (Vaswani et al., 2017), which itself could not deal with sequential information but rely on positional encoding function to map discrete position indexes into differentiable functional domain. Analogically, we need a time encoding function $\mathcal{K}$ for our node-level time attention mechanism, which could map every observed time interval of node connections into a continuous differentiable functional domain, i.e., $\mathcal{K} : [t_e - l, t_e] \to \mathbb{R}^d$. The intuition of involving $\mathcal{K}$ is that, suppose node $v_1$ connects with node $v_2$ at edge timestamp $t_e - l$, when we need to represent the node $v_2$ at edge timestamp $t_e$, we wish the time-aware node representation $\mathbf{u}_{v_2}^{(t_e)}$ incorporates the temporal relationship $\mathcal{K}(t_e - l, t_e)$. This time function $\mathcal{K}$ could reflect the temporal relationship between $\mathbf{u}_{v_1}^{(t_e-l)}$ and $\mathbf{u}_{v_2}^{(t_e)}$, and many previous work solve $\mathcal{K}$ with the kernel method (Zhou et al., 2013; Du et al., 2016; Xu et al., 2019; 2020; Zhang et al., 2020). For example, in (Xu et al., 2019; 2020),

$$\mathcal{K}(t_e - l, t_e) = \Psi(t_e - (t_e - l)) \tag{2}$$

and

$$\Psi(l) = \sqrt{\frac{1}{d}}[\cos\omega_1(l), \cos\omega_2(l), \ldots, \cos\omega_d(l)] \tag{3}$$

where $l = t_e - (t_e - l)$ denotes the input time interval, and $\{\omega_1, \ldots, \omega_d\}$ are learnable parameters.

After sampling the node time-aware adjacent node sequence $N_v^{(t_e)}$ for node $v$, we next apply the node-level time attention on $N_v^{(t_e)}$ to learn $\mathbf{u}_v^{(t_e)}$ by setting node $v$ as the query node to query and aggregate attention weights from previously connected nodes in $N_v^{(t_e)}$. Similar with self-attention mechanism (Vaswani et al., 2017), we need form queries $\mathbf{Q}$, keys $\mathbf{K}$ and values $\mathbf{V}$, then the time-aware node representation $\mathbf{u}_v^{(t_e)} \in \mathbb{R}^r$ can be computed as follows.

$$\mathbf{u}_v^{(t_e)} = \text{Attention}(\mathbf{Q}, \mathbf{K}, \mathbf{V}) = \text{softmax}(\frac{\mathbf{Q}\mathbf{K}^\top}{\sqrt{r}})\mathbf{V} \in \mathbb{R}^r \tag{4}$$

where $\mathbf{Q} = [\mathbf{X}(v, :)\|\mathcal{K}(t_e, t_e)] \cdot \mathbf{W}_Q$, $\mathbf{K} = \mathbf{N} \cdot \mathbf{W}_K$, and $\mathbf{V} = \mathbf{N} \cdot \mathbf{W}_V$. $\mathbf{N} \in \mathbb{R}^{|N_v^{(t_e)}| \times (m+d)}$ is the matrix whose rows are $[\mathbf{X}(v', :)\|\mathcal{K}(t, t_e)] \in \mathbb{R}^{(m+d)}$ from the sequence $N_v^{(t_e)}$, and $\mathbf{W}_Q, \mathbf{W}_K, \mathbf{W}_V \in \mathbb{R}^{(m+d) \times r}$ are three learnable weight matrices with $r$ denoting the dimension of the time-aware node presentation vector $\mathbf{u}_v^{(t_e)}$.

**Intra-Snapshot Time Attention**. After we learn the time-aware node embedding $\mathbf{u}_v^{(t_e)}$ that follows the streaming pattern based at the edge timestamp $t_e$, we also want the node in the snapshot $\mathcal{S}^{(t_s)}$ follows the snapshot pattern of $\mathbf{A}^{(t_s)}$ w.r.t snapshot timestamp $t_s$. Therefore, intra-snapshot time attention is proposed to add constraints on node embeddings $\mathbf{u}_v^{(t_e)}$ in terms of $t_s$ timestamp by reconstructing $\mathbf{A}^{(t_s)}$ via a graph autoencoder.

First, we construct the snapshot feature matrix $\mathbf{U}^{(t_s)} \in \mathbb{R}^{|V^{(t_s)}| \times r}$ whose rows are time-aware node embedding vectors. Note that, snapshot $\mathcal{S}^{(t_s)}$ may not have all nodes of the input temporal graph. For example, for the timestamped edge $(v_1, v_2, t_e, t_s)$, $\mathbf{U}^{(t_s)}(v_1, :) = \mathbf{u}_{v_1}^{(t_e)}$ and $\mathbf{U}^{(t_s)}(v_2, :) = \mathbf{u}_{v_2}^{(t_e)}$. Interestingly, if there is also another edge $(v_1, v_3, t'_e, t_s)$, we will sample the most recent edge timestamp. For example, if $t'_e > t_e$, then $\mathbf{U}^{(t_s)}(v_1, :) = \mathbf{u}_{v_1}^{(t'_e)}$. The reason we adopt the latest node embedding is that, according to the sampling strategy shown in Alg.1, the latest node embedding will encode early node embeddings.

Then, with the adjacency matrix $\mathbf{A}^{(t_s)}$ and snapshot feature matrix $\mathbf{U}^{(t_s)}$, we add a reconstruction loss to learn latent intra-snapshot representation matrix $\mathbf{H}^{(t_s)}$ via the graph autoencoder model (Kipf & Welling, 2016). The snapshot reconstruction loss $\ell$ of the snapshot $\mathcal{S}^{(t_s)}$ is defined as follows.

$$\ell(\mathbf{A}^{(t_s)}, \mathbf{U}^{(t_s)}) = \|\mathbf{A}^{(t_s)} - \hat{\mathbf{A}}^{(t_s)}\|_F \tag{5}$$

where $\hat{\mathbf{A}}^{(t_s)} = \text{GNN}_{dec}(\mathbf{H}^{(t_s)})\text{GNN}_{dec}^{\top}(\mathbf{H}^{(t_s)})$ is the reconstructed adjacency matrix computed as inner product of $\text{GNN}_{dec}(\mathbf{H}^{(t_s)})$ and its transpose, $\mathbf{H}^{(t_s)} = \text{GNN}_{enc}(\mathbf{A}^{(t_s)}, \mathbf{U}^{(t_s)}) \in \mathbb{R}^{|V^{(t_s)}| \times q}$ denotes the intra-snapshot representation matrix, and $\|\cdot\|_F$ denotes the Frobenius norm. $\text{GNN}_{enc}$ and $\text{GNN}_{dec}$ are realized by GCN (Kipf & Welling, 2017) and Sigmoid function, respectively.

Given the extracted intra-snapshot representation matrix $\mathbf{H}^{(t_s)}$, we apply a Readout function to get the intra-snapshot representation vector $\mathbf{h}^{(t_s)}$ at each snapshot timestmap $t_s$ as follows.

$$\mathbf{h}^{(t_s)} = \text{Readout}(\mathbf{H}^{(t_s)}(v, :) \mid v \in \{1, \ldots, |V^{(t_s)}|\} \in \mathbb{R}^q) \tag{6}$$

where Readout is a permutation-invariant function and could be instanced by many graph pooling layer models, such like Zhang et al. (2018); Ying et al. (2018).

**Inter-Snapshot Time Attention**. After we obtain the intra-snapshot representation vector $\mathbf{h}^{(t_s)}$ for each snapshot timestamp $t_s$ individually, we are not sure which one or ones should represent the temporal graph representation vector $\mathbf{z}$ to make it class-distinctive. To be specific, if a certain snapshot $\mathcal{S}$ is shared by different classes of temporal graphs, then that snapshot is less representative and we should decrease its weight during the snapshots aggregation process, to make different class temporal graph representations different.

Therefore, we design a inter-snapshot time attention mechanism on the extracted intra-snapshot representation vectors $\mathbf{h}^{(t_s)}$ to obtain time attention weights for the final temporal graph representation vector $\mathbf{z} \in \mathbb{R}^f$. To be specific, the inter-snapshot time attention is realized through an attention pooling layer (Bahdanau et al., 2015) to get the attention weight $\mathbf{W}^{(t_s)} \in \mathbb{R}^{f \times q}$ for each time $t_s$. Then, the inter-snapshot time attention is parameterized by the learned weight $\mathbf{W}^{(t_s)}$ as follows.

$$\mathbf{z} = \sum_{t_s=0}^{T_s} (\mathbf{W}^{(t_s)} \mathbf{h}^{(t_s)}) \in \mathbb{R}^f \tag{7}$$

### 4.1.2 PROTOTYPE GENERATOR

Through the proposed prototypical temporal graph encoder, we could embed a temporal graph $\mathcal{G}$ into a representation $\mathbf{z}$ as shown in Figure 1. To make same class temporal graphs closer and different class graphs farther apart in the metric $\mathbb{D}$, we need to make same class graph representations closer to their own class prototype and farther from other class prototypes in the metric $\mathbb{D}'$ (i.e., Euclidean).

To this end, in each graph metric learning task $\mathcal{T}_i$, we set the support set $\widetilde{\mathcal{G}}_{support}^{train}$ and the query set $\widetilde{\mathcal{G}}_{query}^{train}$. The prototype learned on $\widetilde{\mathcal{G}}_{support}^{train}$ are used for predicting the class label of graphs in $\widetilde{\mathcal{G}}_{query}^{train}$. The prototype $\mathbf{p}_k$ of the class $k$ is expressed as follows.

$$\mathbf{p}_k = \frac{1}{C_k} \sum_{j}^{C_k} (\mathbf{z}_j) \, , \, \mathcal{G}_j \in \widetilde{\mathcal{G}}_{support}^{train} \text{ and } y_j = k \tag{8}$$

where $\mathcal{G}_j$ is the temporal graph with label $y_j$, $\mathbf{z}_j$ is the embedding of $\mathcal{G}_j$ extracted by the prototypical temporal graph encoder, and $C_k$ denotes the number of $k$ class temporal graphs in $\widetilde{\mathcal{G}}_{support}^{train}$.

---

**Algorithm 2** Meta-Training Process of METATAG

---

**Input:** graph metric learning task distribution $P(\mathcal{T})$, step size hyperparameters $\alpha$ and $\beta$, loss balancing hyperparameter $\gamma$

1:  Randomly initialize $\Theta$  ▷ $\Theta$ denotes all parameters in the prototypical temporal graph encoder
2:  **while** not done **do**
3:      Sample task $\mathcal{T}_i \sim P(\mathcal{T})$ with support set $\widetilde{\mathcal{G}}_{support}^{train}$ and query set $\widetilde{\mathcal{G}}_{query}^{train}$
4:      **for** support set of each task $\mathcal{T}_i$ **do**
5:          Compute $\mathbf{z}_j = f_\Theta(\mathcal{G}_j)$  ▷ $\mathcal{G}_j \in \widetilde{\mathcal{G}}_{support}^{train}$
6:          Construct $\mathbf{p}_k$ for each class $k$ in $\widetilde{\mathcal{G}}_{support}^{train}$ according to Eq. 8.
7:          Evaluate snapshot reconstruction loss $\nabla_\Theta \ell_{\mathcal{T}_i}(f_\Theta)$ and classification loss $\nabla_\Theta \mathcal{L}_{\mathcal{T}_i}(f_\Theta)$
8:          Compute parameter $\theta_i \leftarrow \Theta - \alpha(\nabla_\Theta \ell_{\mathcal{T}_i}(f_\Theta) + \gamma \nabla_\Theta \mathcal{L}_{\mathcal{T}_i}(f_\Theta))$
9:      **end for**
10:     Update $\Theta \leftarrow \Theta - \beta \nabla_\Theta \sum_{\mathcal{T}_i} (\ell_{\mathcal{T}_i}(f_{\theta_i}) + \gamma \mathcal{L}_{\mathcal{T}_i}(f_{\theta_i}))$  ▷ On query set $\widetilde{\mathcal{G}}_{query}^{train}$
11: **end while**

---

To help the the class prototype distinctive to each other, we design the temporal graph classification loss $\mathcal{L}$ in each graph metric learning task $\mathcal{T}_i$ to tune $\theta_i$.

$$\mathcal{L} = - \sum_j^{C_k} \log \frac{\exp(-dist(\mathbf{z}_j, \mathbf{p}_k))}{\sum_{k'} \exp(-dist(\mathbf{z}_j, \mathbf{p}_{k'}))}, \tag{9}$$
$$\mathcal{G}_j \in \widetilde{\mathcal{G}}_{query}^{train} \text{ and } y_j = k$$

where $\mathbf{p}_k$ denotes the $k$ class prototype learned from $\widetilde{\mathcal{G}}_{support}^{train}$, $k'$ denote the class other than $k$, $dist(\cdot)$ denotes Euclidean distance between two vectors, $\mathbf{z}_j$ is the representation vector of $\mathcal{G}_j$ and $C_k$ denotes the number of $k$ class temporal graphs in the set $\widetilde{\mathcal{G}}_{query}^{train}$.

## 4.2 META-LEARNER

We have introduced the whole learning procedure and two loss functions (i.e., Eq. 5 and Eq. 9) for extracting knowledge $\theta_i$ from a single task $\mathcal{T}_i$. Next, we need to break though the knowledge transfer and adaption cross tasks given only few-shot examples. Here, we introduce a meta-learner to transfer the learned knowledge $\theta_i$ and tailor the globally shared knowledge $\Theta$, the theory behind is that transferring shareable knowledge could obtain the fast convergence on unseen tasks (Chauhan et al., 2020; Ma et al., 2020).

We formalize the meta-training process of METATAG in a bi-level paradigm (Finn et al., 2017), which is able to find meta-learner $\Theta$ that could be fast converged in each graph metric learning task. As shown in Algorithm 2, we first randomly initialize $\Theta$ in Step 1. Then, in each graph metric learning task $\mathcal{T}_i \sim P(\mathcal{T})$, we obtain the temporal graph representation vector in Step 5 and build the class prototype for each class of the support set in Step 6. In Step 8, we tune $\Theta$ to get $\theta_i$ for current task $\mathcal{T}_i$. In Step 10, we aggregate the loss from each task $\mathcal{T}_i$ and fine tune $\Theta$ to end the meta-training process. After that, we can use the fine-tuned $\Theta$ as the initialized parameter in meta-testing stage for unseen graph metric learning tasks, aiming to the fast adaptation via only a few labeled samples.

The meta-testing phase of METATAG is very similar to Algorithm 2. After changing $\widetilde{\mathcal{G}}_{support}^{train}$ into $\widetilde{\mathcal{G}}_{support}^{test}$ and $\widetilde{\mathcal{G}}_{query}^{train}$ into $\widetilde{\mathcal{G}}_{query}^{test}$, the only difference is that Step 10 directly reports the accuracy based on $\theta_i$ instead of getting new $\Theta$.

## 5 EXPERIMENTS

In this section, we test our METATAG in terms of temporal graph classifications comparing with state-of-the-art graph kernel and graph metric learning baseline algorithms. More experimental details about the implementation and the other extensive experimental results like convergence speed, parameter sensitivity, and ablation study can be found in Appendix.

## 5.1 EXPERIMENT SETUP

**Datasets**. Our experiments conclude 12 temporal graph datasets from the biological domain (Fu & He, 2021), and 6 temporal graph datasets from the social network domain (Morris et al., 2020). Each biological graph is a dynamic protein-protein interaction network, which describes the proteins interact of metabolic cycles of different yeast cells, where each node stands for a protein, and timestamped edge stands for the interact of a pair of proteins. Each social network is a human-contact relation graph online and offline, where the edges between individuals stand for the online or offline contacts. The statistics of all network data are summarized in Table 1 and Table 2.

Table 1: Statistics of Biological Temporal Graph Data

| Graph | #Classes | #Graphs | Total Nodes | Total Edges | Timestamps | Graph | #Classes | #Graphs | Total Nodes | Total Edges | Timestamps |
|---|---|---|---|---|---|---|---|---|---|---|---|
| Uetz | 1 | 11 | 922 | 2,159 | 36 | Ito | 1 | 11 | 2,856 | 8,638 | 36 |
| Ho | 1 | 11 | 1,548 | 42,220 | 36 | Gavin | 1 | 11 | 2,541 | 140,040 | 36 |
| Krogan-LCMS | 1 | 11 | 2,211 | 85,133 | 36 | Krogan-MALDI | 1 | 11 | 2,099 | 78,297 | 36 |
| Yu | 1 | 11 | 1,163 | 3,602 | 36 | Breitkreutz | 1 | 11 | 869 | 39,250 | 36 |
| Babu | 1 | 11 | 5,003 | 111,466 | 36 | Lambert | 1 | 11 | 697 | 6,654 | 36 |
| Tarassov | 1 | 11 | 1,053 | 4,826 | 36 | Hazbun | 1 | 11 | 143 | 1,959 | 36 |

Table 2: Statistics of Social Temporal Graph Data

| Graph (Online) | #Classes | #Graphs | Total Nodes | Total Edges | Timestamps | Graph (Offline) | #Classes | #Graphs | Total Nodes | Total Edges | Timestamps |
|---|---|---|---|---|---|---|---|---|---|---|---|
| Facebook | 2 | 995 | 95,224 | 267,673 | 104 | Infectious | 2 | 200 | 10,000 | 91,944 | 48 |
| Tumblr | 2 | 373 | 19,811 | 74,520 | 89 | HighSchool | 2 | 180 | 9,418 | 98,066 | 203 |
| DBLP | 2 | 755 | 39,917 | 241,674 | 46 | MIT | 2 | 97 | 1,940 | 142,508 | 5,576 |

**Baselines**. The selection of baseline algorithms includes three factors, i.e., graph kernel or graph metric learning, few-shot learning or not few-shot learning, and static or dynamic. Graph kernel methods include: Vertex histogram kernel (Nikolentzos et al., 2019), Shortest Path kernel (Borgwardt & Kriegel, 2005), Neighborhood Hash graph kernel (Hido & Kashima, 2009), Weisfeiler-Lehman Optimal Assignment kernel (Kriege et al., 2016), and Pyramid Match kernel (Nikolentzos et al., 2017). Graph metric learning or graph representation learning algorithms include: GL2Vec (Chen & Koga, 2019), NetLSD (Tsitsulin et al., 2018), tdGraphEmbed (Beladev et al., 2020), TGAT (Xu et al., 2020), and CAW (Wang et al., 2021). GL2Vec and NetLSD are static algorithms, tdGraphEmbed is a dynamic algorithm that could take a temporal graph as input and output graph embeddings of each snapshot, and TGAT and CAW are dynamic graph representation learning algorithms but focus on the node-level. To enable graph metric learning methods the few-shot learning capability, we also include ProtoNet (Snell et al., 2017) and its special case k-NN method.

Table 3: Temporal Graph Classification Accuracy on Biological Temporal Graphs

| | Methods | 3 way - 5 shot | 3 way - 3 shot | 3 way - 2 shot | 3 way - 1 shot |
|---|---|---|---|---|---|
| Graph Kernel | Weisfeiler-Lehman Opt | $0.5025 \pm 0.3531$ | $0.4625 \pm 0.3118$ | $0.4350 \pm 0.2420$ | $0.4250 \pm 0.2251$ |
| | Vertex Histogram | $0.3150 \pm 0.2466$ | $0.2700 \pm 0.1881$ | $0.1375 \pm 0.1314$ | $0.3125 \pm 0.2415$ |
| | Neighborhood Hash | $0.4375 \pm 0.4058$ | $0.4400 \pm 0.3697$ | $0.2850 \pm 0.1815$ | $0.4000 \pm 0.3175$ |
| | Pyramid Match | $0.2500 \pm 0.1971$ | $0.2525 \pm 0.1337$ | $0.2325 \pm 0.1569$ | $0.2950 \pm 0.2174$ |
| | Shortest Path | $0.2025 \pm 0.1477$ | $0.2175 \pm 0.1314$ | $0.1875 \pm 0.1325$ | $0.1900 \pm 0.1329$ |
| Graph Metric Learning | GL2Vec + KNN | $0.1400 \pm 0.0616$ | $0.1925 \pm 0.0754$ | $0.1175 \pm 0.0689$ | $0.1150 \pm 0.0591$ |
| | NetLSD + KNN | $0.3600 \pm 0.2585$ | $0.3650 \pm 0.2747$ | $0.2000 \pm 0.0901$ | $0.2625 \pm 0.1519$ |
| | TGAT + KNN | $0.2100 \pm 0.0817$ | $0.1325 \pm 0.2217$ | $0.1650 \pm 0.0387$ | $0.0750 \pm 0.0208$ |
| | tdGraphEmbed + KNN | $0.3200 \pm 0.1272$ | $0.2275 \pm 0.1459$ | $0.1750 \pm 0.0580$ | $0.1875 \pm 0.0150$ |
| | GL2Vec + ProtoNet | $0.6083 \pm 0.0099$ | $0.6541 \pm 0.0159$ | $0.6542 \pm 0.1370$ | $0.5583 \pm 0.1578$ |
| | NetLSD + ProtoNet | $0.6916 \pm 0.1396$ | $0.7145 \pm 0.1396$ | $0.6937 \pm 0.1674$ | $0.6667 \pm 0.1372$ |
| | TGAT + ProtoNet | $0.2417 \pm 0.0500$ | $0.3083 \pm 0.0739$ | $0.2917 \pm 0.1167$ | $0.2417 \pm 0.0319$ |
| | CAW + ProtoNet | $0.1496 \pm 0.0104$ | $0.2113 \pm 0.0110$ | $0.2404 \pm 0.0117$ | $0.2842 \pm 0.0044$ |
| | tdGraphEmbed + ProtoNet | $0.6562 \pm 0.1882$ | $0.6791 \pm 0.1141$ | $0.6271 \pm 0.1159$ | $0.4229 \pm 0.0463$ |
| | MetaTag (Ours) | $\mathbf{0.7292 \pm 0.0682}$ | $\mathbf{0.7917 \pm 0.1278}$ | $\mathbf{0.7062 \pm 0.0762}$ | $\mathbf{0.6833 \pm 0.0589}$ |

## 5.2 TEMPORAL GRAPH CLASSIFICATION

First, in the biological dataset, given the 12 classes we split 8 classes into the meta-training set $\widetilde{\mathcal{G}}^{train}$ and 4 classes into the meta-testing test $\widetilde{\mathcal{G}}^{test}$. Note that $\widetilde{\mathcal{G}}^{train}$ and $\widetilde{\mathcal{G}}^{test}$ do not share any class label. In $\widetilde{\mathcal{G}}^{train}$, we sample $K$-way $N$-shot graph metric learning tasks $\mathcal{T}_i \sim P(\mathcal{T})$, and each task has a support set $\widetilde{\mathcal{G}}^{train}_{support}$ and a query set $\widetilde{\mathcal{G}}^{train}_{query}$. Then, METATAG is trained on $\widetilde{\mathcal{G}}^{train}$

based on Algorithm 2 and fine tune a few times on $\widetilde{\mathcal{G}}_{support}^{train}$ and report the accuracy on $\widetilde{\mathcal{G}}_{query}^{train}$. We shuffle $\widetilde{\mathcal{G}}^{train}$ and $\widetilde{\mathcal{G}}^{test}$ 4 times for cross-validation and report the average classification accuracy in Table 3, where our METATAG outperforms all the baseline algorithms. For example, in the 3-way 5-shot setting, our METATAG achieve $72.92\%$ temporal graph classification accuracy, which is $5.44\%$ higher than the second place. An intuitive explanation is that different class yeast cells (i.e., temporal graphs) has class-distinctive metabolic patterns (i.e., temporal patterns), capturing that pattern comprehensively is helpful in identifying class labels. Also, we observe other interesting patterns. First, graph kernel methods and KNN-based graph metric learning methods do not perform well and bear the larger standard deviation. A possible answer is that they do not have few-shot learning capability and could not transfer knowledge from seen cases to unseen cases. ProtoNet-based graph metric learning and our method enjoy the data augmentation property from the few-shot learning manner, thus they have better performance and smaller deviations. Second, our experiments shows that increasing the number of shots during the meta-training is not always the good choice for improving the performance of meta-testing [1], because intra-class variances may be amplified (Cao et al., 2020). For the page limit, we place the experimental results of temporal graph classification on social network data in Appendix.

## 6 RELATED WORK

**Graph Metric Learning**. Learning a good metric in the input feature can be transferred to learn proper graph representations in Euclidean space, then graph embedding based graph metric learning methods are proposed (Shaw et al., 2011; Bai et al., 2019; Li et al., 2019). Facing the label scarcity problem, many generic metric learning methods consider the few-shot learning or meta-learning strategy to adapt metrics across different tasks with only a few labeled sample in each task (Snell et al., 2017; Oreshkin et al., 2018; Allen et al., 2019). Inspired by that, some graph metric learning methods involve the few-shot learning manner, where the majority of these algorithms focus on learning the metric over nodes across different graphs (Yao et al., 2020; Suo et al., 2020; Huang & Zitnik, 2020; Lan et al., 2020; Wang et al., 2020; Ding et al., 2020). Only a few graph metric learning methods learn the metric over the whole graphs to distinguish distance between graphs (Ma et al., 2020; Chauhan et al., 2020). Currently, graph metric few-shot learning methods ignore to consider the dynamics of graphs into the metric learning process. We are the first effort to involve the dynamics and temporal dependencies of input graphs into the learned metric. **Graph Kernel**. Given a distance metric $\mathbb{D}$, it should maintain four properties: non-negativity (i.e., $\mathbb{D}(\mathbf{x}, \mathbf{y}) \geq 0$), coincidence (i.e., $\mathbb{D}(\mathbf{x}, \mathbf{y}) = 0$ iff $\mathbf{x} = \mathbf{y}$), symmetry (i.e., $\mathbb{D}(\mathbf{x}, \mathbf{y}) = \mathbb{D}(\mathbf{y}, \mathbf{x})$), and subadditivity (i.e., $\mathbb{D}(\mathbf{x}, \mathbf{y}) + \mathbb{D}(\mathbf{y}, \mathbf{z}) \geq \mathbb{D}(\mathbf{x}, \mathbf{z})$ ) (Wang & Sun, 2015). While in graph kernel research, only the symmetry and non-negativity need to be hold for a kernel function, i.e., the symmetric graph kernel function $\mathcal{K}$ should be a positive semi-definite function (Vishwanathan et al., 2010). To measure the similarity among graphs, one category graph kernel methods explicitly define the kernel function from the graph topological view, such as Random Walk graph kernel (Vishwanathan et al., 2010) and Weisfeiler-Lehman graph kernel (Shervashidze et al., 2011). To handle the evolving graph scenario, some methods map dynamic graphs into constant representations and then apply static graph kernel functions for dynamic node classification (Yao & Holder, 2014) and temporal graph classification (Oettershagen et al., 2020); On the other hand, some graph kernel methods learn the kernel function instead of hand-crafted designing it (Yanardag & Vishwanathan, 2015; Zhao & Wang, 2019). For example, in (Yanardag & Vishwanathan, 2015), the kernel is determined by learning the latent representation of substructures of input graphs.

## 7 CONCLUSION

In this paper, we first propose the streaming-snapshot model to describe a temporal graph, and then we propose the prototypical temporal graph encoder to capture temporal graph representation vectors. Last but not the least, we entitle the prototypical temporal graph encoder with a meta-learner into an end-to-end model, named METATAG, to transfer knowledge among different tasks for the fast adaption to unseen cases. We execute extensive experiments to show the effectiveness of our METATAG with different category state-of-the-art baseline algorithms.

---

[1]The number of shots in the query set during meta-testing is 2.

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
