# OpenReview forum: "Metric Learning on Temporal Graphs via Few-Shot Examples"
_ICLR.cc/2022/Conference — ICLR 2022 Submitted_

### Official Review · Reviewer_Qtrm · 2021-10-29

**Correctness:** 3
**Technical Novelty And Significance:** 2
**Empirical Novelty And Significance:** 2
**Recommendation:** 5
**Confidence:** 3

**Main Review:**

The definition and example of the temporal graph are confusing. I could see that that a temporal graph consists of snapshots. I could not see why in snapshot, there are edge timestamps? Do you assume the node set fixed? Do you mean that a snapshot is also a temporal graph with edge appear and disappear at different times (a)? If yes, then I could not see it in the Figure 2. I could not see the meaning of t_e=0/1/2 in the figure.

I'd be assuming (a), then the statements like "previous methods only focus on one time scale and ignore the whole lifetime evolution representation" are difficult to understand. I don't see why methods for temporal graphs do not take all the times of the graphs into account, i.e. do not take the whole graphs into account.

The method used in the paper is a combination of different standard methods available. There is nothing wrong with that, but it would be useful to prove the usefulness for each of these choices.

Overall, I could see that the proposed method is supposed to be general with multi-scale timestamps and claim that, due to its generality, it is better than all other methods available out there. This kind of messages, in my opinion, needs to be more specific on when/why.

**Summary Of The Paper:**

The paper proposed a method for metric learning for few-short examples, where each example is a temporal graph modeled with two timescales.

**Summary Of The Review:**

The problem is not clearly defined and hard to read. The method is general and claimed to be better than others due to its generality. I do not see too much novelty in the method. The results look good.

---

### Official Review · Reviewer_5yGt · 2021-11-02

**Correctness:** 3
**Technical Novelty And Significance:** 2
**Empirical Novelty And Significance:** 2
**Recommendation:** 3
**Confidence:** 3

**Main Review:**


The strengths are:

1. The problem considered in this paper is interesting and new. Specifically, the authors consider the graph metric learning but in the temporal setting.

2. The authors proposed a new method, which includes two parts. To capture the temporal information, the key of the proposed is based on three different attentions. Previous works such as Xu et al, 2020 and Yang et. al., 2021 have demonstrated the power of time-related attention layers.

3. The temporal graph classification experiments on two datasets demonstrate that the proposed method learns much better classifiers than other baseline methods.

The weaknesses are:

1. The contribution of the streaming-snapshot model is not critically novel. The essential idea of the streaming-snapshot model is to combine the discrete-time dynamic graph (DTDG) and continue the time dynamic graph (CTDG) model together (see more details in [1]). The authors use both two models but may have different time granularity.

2. The technical contributions are limited. The whole framework may look interesting and new but it is based on Finn et. al., 2017. The time attention layers are commonly used in temporal graphs which are also proposed in previous work.

3. The experiments are limited. First of all, the run time analysis is missing from both theoretical and empirical perspectives. More explanations are needed on the results of GL2Vec + ProtoNet. The experimental setup and parameter settings are unclear.

---
Minors:
1. It is a little bit unfair to directly use CAW and TGAT under the graph metric learning setting. What is the specific loss used for these two?
2. Table 3, MetaTag → \textsc{MetaTag}
3. Why directly adding ProtoNet onto CAW and TGAT be a reasonable thing?
4. I expect detailed experimental setups like parameter tuning in the appendix. Could you list the experimental details in the appendix?
5. What is the scalability of this method compared with two strong baselines: 1) GL2Vec + ProtoNet and 2) tdGraphEmbed + ProtoNet? It seems that GL2Vec + ProtoNet is competitive on the social network datasets but bad on the bio-based datasets. Why is this case?

[1] Kazemi SM, Goel R, Jain K, Kobyzev I, Sethi A, Forsyth P, Poupart P. Representation Learning for Dynamic Graphs: A Survey. J. Mach. Learn. Res.. 2020 Jan 1;21(70):1-73.



**Summary Of The Paper:**

This paper considers the graph metric learning problem where the underlying graphs are temporal. The key idea of obtaining a higher classification accuracy is to use a bi-level meta-learning paradigm. It essential contains two parts: 1) prototypical temporal graph encoder where the model uses multi-scale time attention to capture temporal information; and 2) meta-learner where it uses bi-level paradigm proposed in Finn et al., 2017. The authors apply the proposed method to the task of graph classification on two real-world datasets. Compared with other baseline methods, MetaTag achieves better performance in terms of classification accuracy.

**Summary Of The Review:**

In general, it is an interesting paper. The experimental results look promising. However, the overall quality is not strong enough for acceptance. There is space for improvement in experiments. For example, the comparison between the proposed method and baseline methods. More discussion on the experimental results is also needed.

---

### Official Review · Reviewer_jL2u · 2021-11-02

**Correctness:** 2
**Technical Novelty And Significance:** 2
**Empirical Novelty And Significance:** 2
**Recommendation:** 3
**Confidence:** 4

**Main Review:**


# Strengths
- First attempt to provide an end-to-end differentiable model for handling time-varying graphs
- Addresses separate time-scales (using streaming/snapshot model)
- Adapts and uses existing methodology where appropriate (GCNs/Readout) for snapshot representation, ProtoNets for few-shot learning, learnable time-kernels and attention for aggregating the time-aware influence of neighbouring node features into node representation.

# Weaknesses
1. The paper needs some rewriting starting with separate notions of time (streaming/snapshot) - classic stochastic theory handles this by consider $t_e$ as discrete time and ${\cal S}^{(t)} |_{t=t_s}$ is the observation event (when we see the interactions).  As described by the authors, the snapshots model episodic, slow-changing, and periodical patterns. However, the model does not correspond to "time elapsed" between consecutive snapshots $(s-1), s, (s+1)$ beyond ordering. Further, it is not clear how to relate $t_s$ to $t_e$ (which is the 'real time').  I would recommend the authors call it "k" snapshots and not call it snapshot 'time' for clarity.

2. The attention weighting across snapshots is agnostic to "elapsed time" meaning that few-shot learning results will work well if the time between consecutive snapshots is same across original task and new task (see point 3 below as well).


3. The biological dataset [DPPIN, Fu and He, 2021] is not 12 separate time-varying graphs rather one large graph and a _single_ time-varying gene expression dataset from which dynamics are inferred over twelve different subgraphs. This means in the fine-tuning setting, the learnt weighting (equation 7) benefits from the fact that the temporal dynamics are representing the same underlying time-scale (snapshots in test and train correspond to real gene expression values in the same experiment at the same instant) which is why the attention weighting works well. I would be very interested in seeing how few-shot learning works when considering two different gene expression arrays for deriving the underlying dynamics.

4. Would it make sense to re-order Algorithm 1 (lines 1-2) in terms of latest edge connecting to a node for a snapshot, i.e., to generate ${\bf U}^{(t_s)}$, for each node $v$ select latest edge $(v, v', t')$ and use time-aware attention mechanism to get influence of pre-existing edges $(v, v'', t''), t'' \le t'$  into its node representation. From a computational perspective, is only ${\bf u}^{t'}_v$ computed or all ${\bf u}^{t''}_v$ computed first and only the latest one selected in  ${\bf U}^{(t_s)}$.

5. (a) When computing ${\bf U}^{(t_s)}$, suppose you have only the following edges $(v_1, v_2, 0)$, $(v_2, v_3, 1)$, $(v_3, v_4, 2)$ would the final ${\bf U} = [ u_1^0, u_2^1, u_3^2, u_4^2]$?
    (b) In Algorithm 1. Should $t < t_e$ be $t \le t_e$ otherwise consider two edges $(x, y, t_e=3)$  and $(x, y', t_e=3)$ - then the influence of the other edge is missed. Further, how is this addressed in ${\bf U}^{(t_s)}$.
    An example in the text would clarify here.

6. In the appendix A.2, the lines "For each temporal graph, 36 edge timestamps together describe three consecutive metabolic cycles. In each graph, we take a subgraph by extracting a interval of 5 edge timestamps every 3 edge timestamps. The subgraph shares the same class label with its original entire graph. Therefore, we have 11 temporal subgraphs per class." are directly copied from https://arxiv.org/pdf/2107.02168.pdf  "in each graph, we take a subgraph by extracting the time interval of five timestamps every three timestamps. The subgraph shares the same class label with its original entire graph. Therefore, we have eleven temporal subgraphs per class." It is not clear to me how eleven (and not twelve) temporal subgraphs are obtained. Further, there is a mismatch in the Table 1 (#classes shown as 1) and section 5.2 Line 1 (given the 12 classes). The description of the dataset should be expanded upon and made consistent.

# Minor corrections
- Equation 1. ${\bf y}_j$ should be ${\bf x}_j$
- In section 3, the authors should define a time-stamped edge $(v_i, v_j, t_e, t_s)$. My understanding is that this means an edge between nodes $v_i$ and $v_j$ that formed at "microscopic" time scale $t_e$ and is present whenever the $t_s^{th}$ snapshot was taken (clearly sometime after $\max \{ t_e | (v_i, v_j, t_e, t_s) \in S^{(t_s)} \}$).

- Page 9, Line 1: "and fine tune a few times on $\tilde{G}^{train}\_{support}$  and report the accuracy on  $\tilde{G}^{train}\_{query}$". Should this be "fine tune a few times on $\tilde{G}^{test}\_{support}$ and report the accuracy on  $\tilde{G}^{test}\_{query}$"?
- On page 3, it should be made clear whether ${\bf A}^{(t_s)}$ the snapshot adjacency matrix is just the presence of an edge at that instant (regardless of when it was created) i.e., $(i, j, t_e) \in E^{(t_{s})} \Rightarrow A_{ij}^{(t_s)}=1$.



**Summary Of The Paper:**

This authors present a novel method for learning representations for time-varying graphs which allows for incorporating information at different time-scales using their streaming-snapshot model. The streaming-snapshot model has the following parts:
* Each snapshot $S^{(t_{s})} = (V^{(t_{s})}, E^{(t_{s})})$ has edges of the form $(v_i, v_j, t_e) \in E^{(t_{s})}$ where $t_e$ denotes the time at which edge was formed (and is present since then).
* The snapshots $S^{(t_{s})}$ are at a different time-scale ($t_e$ and $t_s$ are not comparable) with the overall learning representation being ${\cal S} \to \Re^f$.
* Learning this representation is used for downstream few-shot classification task (for dynamic graphs) and is evaluated on two scenarios - time-varying biological (protein-protein interaction) networks and time-varying social networks.

The MetaTag architecture has the following components:

* _Time-aware node representation_ The edge creation time $t_e$ is used to learn a time-aware node representation ${\bf u}^t_e$ using attention-based weighting of neighbouring nodes features concatenated with a learnable time kernel (Algorithm 1).
- The snapshot feature matrix $U^{(t_{s})}$ takes the node representation by consider the latest edge for node $u$ and using attention mechanism above to get influence of earlier edges.
* _Intra-snapshot representation_ This is constructed using standard representation loss using a GCN-based encoder-decoder architecture followed by permutation-invariant readout to obtain vector representation for snapshot.
* _Overall representation_ The overall representation for the time-varying graph is  weighted average using attention pooling (learnt parameter) of different snapshot representations.

This representation is used downstream for classification task (classification-head) based on prototypical approach [Snell+, 2017] resulting in overall end-to-end differentiable model with weighted average of reconstruction loss and classification loss. Further, the model allows adaptation to new tasks (with different classification labels) by fine-tuning on small test set (few-shot learning).

Experiments are shown on biological and social network datasets (in appendix) showing efficacy of the approach compared to static graph representation methods as well as tdGraphEmbed (doc2vec style method for embedding temporal graphs) including augmentation with ProtoNet for few-shot learning comparison.


**Summary Of The Review:**

The current paper addresses an important problem using a smart approach but requires significant rewriting as well as better empirical evaluation. Results on social network data are only marginally better than a much simpler approach (GL2Vec+ProtoNet) while the biological time-varying graph dataset is not actually different tasks since the underlying dynamics for all the 12 networks are learnt from a single gene-expression dataset which means train and test settings share the same underlying biological process and time-scale. I do not believe the paper is ready for publication in the present form.

---

### Official Review · Reviewer_sLfA · 2021-11-02

**Correctness:** 4
**Technical Novelty And Significance:** 2
**Empirical Novelty And Significance:** 2
**Recommendation:** 5
**Confidence:** 3

**Main Review:**

**Strengths:**

- The idea of enriching a graph representation using knowledge of its temporal dynamics is quite nice. The authors split the time attention into three parts: node-level, intra-snapshot, and inter-snapshot. The node-level attention mechanism supplies temporal sensitivity to the node representations, the intra-snapshot mechanism adds in a graph autoencoder to learn how to reconstruct the adjacency matrix of a snapshot, and the inter-snapshot attention mechanism weights different snapshots according to discriminative power. The modular design is helpful in understanding how different aspects of the pipeline can be improved in future work.

- The additional meta-learning module is a nice addition, and it is useful to see how to interface this module with learning dynamic graph representations.


**Weaknesses/points to clarify:**


- The use of $t_e \in T_e$ is unclear from Figure 2. Could the authors please clarify how $t_e$ is obtained? Currently it seems that we have a set of edges at $t_s=0$ with $max_{t_e} = 2$, and that the new edges added at $t_s = 1$ have $t_e=3$, i.e. incremented by 1. Similarly, at $t_s=2$, the new edges are labeled with $t_e=4$. So the figure doesn't shed light into the differences between the two timescales $t_e$ and $t_s$ beyond "increment by 1". Section A.2 contains further mention of $t_e$ but without explanation, so I still don't fully understand how $t_e$ is generated. I think this issue can be fixed with a few lines of clarifying language.

- In Section 4.1.1, where is the attention in the "intra-snapshot time attention" module? It seems that here we just learn how to reconstruct the adjacency matrix at each snapshot. Perhaps this can be clarified in the author response.

- Possibly more seriously, I'm concerned about the **novelty** of the proposed framework. Consider references [A],[B] and the further references contained therein to attention-based models for learning dynamic graph representations. These do not appear as baselines in the current work, and in fact do not even appear as references. It seems that the meta-learning portion in the current work is new, but is that the only source of novelty? I hope the authors will be able to clarify these connections in their response.

[A0] Sankar, A., Wu, Y., Gou, L., Zhang, W., & Yang, H. (2018). Dynamic graph representation learning via self-attention networks. arXiv preprint arXiv:1812.09430.

[A1] Sankar, A., Wu, Y., Gou, L., Zhang, W., & Yang, H. (2020, January). Dysat: Deep neural representation learning on dynamic graphs via self-attention networks. In Proceedings of the 13th International Conference on Web Search and Data Mining (pp. 519-527).

- (A1 is an expanded version of A0)

[B] Rossi, E., Chamberlain, B., Frasca, F., Eynard, D., Monti, F., & Bronstein, M. (2020). Temporal graph networks for deep learning on dynamic graphs. arXiv preprint arXiv:2006.10637.

**Summary Of The Paper:**

This paper introduces a methodology of graph learning for dynamic graphs, where the dynamics are encoded in the representation to obtain improved results on graph classification tasks. This framework includes a temporal graph encoder that uses attention mechanisms to generate representations, as well as a meta-learning component that ensures easy knowledge transfer. Experiments are carried out on two temporal graph datasets to show strong performance in graph classification.

**Summary Of The Review:**

I think the paper has good experimental results and that the incorporation of the meta-learning framework is useful. However, I am concerned that the authors have left out important references, and would welcome a thorough evaluation of these references (and ideally other important references contained therein) and how they compare to the current work. Currently I will lean toward reject, but welcome a discussion with the authors on the novelty of their work.

---

### Decision · Program_Chairs · 2022-01-20

**Decision:**

Reject

**Comment:**

The paper proposes a new method for representation learning of time-varying graphs which uses a streaming-snapshot model to describe graphs on different time scales and meta-learning for adaption to unseen graphs. Reviewers highlighted as strengths that the paper proposes an interesting approach for modeling temporal dynamics in graphs which of interest to the ICLR community. However, reviewers raised also concerns regarding the novelty of contributions, the empirical evaluation (also with regard to related work), as well as the clarity of presentation. In addition there was no author response. All reviewers and the AC agree therefore that the paper is not yet ready for publication at ICLR at this point.